# The relationships between children's motor competence, physical activity, perceived motor competence, physical fitness and weight status in relation to age

**Anne R. den Uil** [1,2]*, **Mirka Janssen**[1], **Vincent Busch**[3], **Ilse T. Kat**[1], **Ron H. J. Scholte**[2]

**1** Centre of Expertise Urban Vitality, Faculty of Sports and Nutrition, Amsterdam University of Applied Sciences, Amsterdam, The Netherlands, **2** Behavioural Science Institute, Radboud University Nijmegen, Nijmegen, The Netherlands, **3** Department Sarphati Amsterdam, Public Health Service (GGD) Amsterdam, Amsterdam, The Netherlands

* a.r.den.uil@hva.nl

## Abstract

The goal of this cross-sectional study was to further explore the relationships between motor competence, physical activity, perceived motor competence, physical fitness and weight status in different age categories of Dutch primary school children. Participants were 2068 children aged 4 to 13 years old, divided over 9 age groups. During physical education classes, they completed the 4-Skills Test, a physical activity questionnaire, versions of the Self-Perception Profile for Children, Eurofit test and anthropometry measurements. Results show that all five factors included in the analyses are related to each other and that a tipping point exists at which relations emerge or strengthen. Physical fitness is related to both motor competence and physical activity and these relationships strengthen with age. A relationship between body mass index and the other four factors emerges in middle childhood. Interestingly, at a young age, motor competence and perceived motor competence are weakly related, but neither one of these have a relation with physical activity. In middle childhood, both motor competence and perceived motor competence are related to physical activity. Our findings show that children in late childhood who have higher perceived motor competence are also more physically active, have higher physical fitness, higher motor competence and lower body mass index. Our results indicate that targeting motor competence at a young age might be a feasible way to ensure continued participation in physical activities throughout childhood and adolescence.

## Introduction

Physical activity (PA) levels have been decreasing worldwide [1]. Of the Dutch population aged 4 and up, 51% does not meet the recommendations for daily physical activity [2]. Physical activity reduces the risk of cardiovascular disease, cancer and type-2 diabetes, improves musculoskeletal health and can reduce symptoms of depression [3]. Also, physical activity is an

**Data Availability Statement:** The data have been uploaded to Figshare with DOI https://doi.org/10.21943/auas.22332841.v1.

**Funding:** This work was supported by a grant for AU from the Netherlands Organization for Scientific Research (NWO) (grant number 023.013.055). https://www.nwo.nl/ The funders had no role in study design, data collection and analysis, decision to publish, or preparation of the manuscript.

**Competing interests:** The authors have declared that no competing interests exist.

important factor influencing weight status, including obesity [4]. Obesity is one of the biggest health challenges around the world [5]. It increases the risk of type-2 diabetes, cardiovascular diseases and several forms of cancer. In addition, it imposes an economic burden on society [6]. In the Netherlands, 43.9% of the population is overweight, including 13.0% of the population that is obese. Of the Dutch 4- to 11-year old children, 11.9% is already overweight [7]. In order to increase levels of physical activity and reduce obesity among children, increased attention is paid to (school-based) interventions. However, the effectiveness of these interventions has been limited. Therefore, using a different approach for promoting physical activity might be useful.

A stronger focus on motor competence in primary school could be such approach. Fundamental motor skills (FMS) are the building blocks of more advanced, complex movements required to participate in sports, games or other context specific physical activity and include object control skills (i.e. throwing and catching), locomotor skills (i.e. hopping, skipping) and balance/stability skills (i.e. one-foot balance, turning) [8]. A recent review [9] of longitudinal studies concludes that there is strong evidence that overall motor competence (but not necessarily specific skills domains) influences physical activity levels. Since an increasing number of children develop motor delays somewhere during primary school [10], improving competence in motor skills of young children might be a feasible focus for improving physical activity levels and weight status.

Although the role of perceived motor competence in the interaction between motor competence and physical activity is often overlooked, a child's perception of its motor competence is possibly even more important for physical activity levels than their actual motor competence [11, 12]. For example, while no direct association was found between motor competence and physical activity in some studies, indirect associations via perceived motor competence and physical self-concept were found [13, 14]. Stodden et al. [15] include both motor competence and perceived motor competence in their developmental model. They state that these are important factors influencing physical activity levels and that either a positive or a negative spiral can develop. In short: if motor competence develops slower than that of peers, children will be aware of that. Their perceived motor competence lowers, making them more likely to drop out of physical activities, influencing weight status, hereby completing a negative spiral. On the other hand, when children develop an adequate competence in motor skills, a positive spiral will develop.

A key aspect of the *developmental* model by Stodden et al. [15] is that it suggests that these relationships change according to the developmental stages of a child. At a young age, physical activity stimulates motor skill development. At this time, children cannot distinguish between actual skills and effort, leading to inflated levels of perceived competence [16]. As a result, children continue to participate despite their actual competences, leading to more opportunities to improve their motor competence. With the development of self-awareness, perceived competence starts to reflect actual motor competence more accurately [17] and this starts to influence physical activity levels. The ongoing development of physical activity and motor skills is thus influenced by perceived motor competence. As a consequence, the spiral of disengagement in physical activity begins in children with low motor competence [15]. Therefore, the timing of an intervention might be a key factor in influencing daily physical activity levels. Motor skills should be developed before perceived motor competence starts to reflect actual motor competence, then hindering physical activity participation. Motor skills should thus be improved at a very young age, especially since there might be a sensitive period for acquiring competence in motor skills [18–20].

An age-dependency of the relationship between perceived motor competence and physical activity has been supported in literature [21]. Although evidence for the mediating role of

perceived motor competence in the relation between motor competence and physical activity is insufficient, based on available studies, age also seems to influence this mediating relationship [9]. Still, the question remains when this perceived motor competence starts to impact children's physical activity behavior [22].

Several factors are thus involved in a child's physical development. In addition to the aforementioned factors, a reciprocal role of health-related physical fitness is also included in the model [15]: at a young age physical activity and motor competence stimulate physical fitness, but as children grow older physical fitness also influences motor competence and physical activity in return [15]. So, weight status, physical activity, motor competence, perceived motor competence and physical fitness might all be interrelated.

So far, research has mainly focused on exploring the separate proposed relationships. Some cross-sectional studies have studied multiple relationships, but this was only done in specific age groups, mainly in older children [12, 23–27]. So although the developmental nature of these relationships is an important aspect of this model, research is mainly focused on separate age categories and the strengths of associations are often not reported [9, 28]. Research including all factors and various age-categories, thus the total model, is missing. Further exploration of these relationships is thus essential. Therefore, in the present cross-sectional study, we aim to further explore the relationships between weight status, physical activity, motor competence, perceived motor competence and physical fitness in Dutch primary school children. Specifically, these relationships are studied in children ranging from 4 to 13 years old, providing more insight into these relationships through developmental time.

## Materials and methods

### Design and setting

This study used a cross-sectional design. Six primary schools participated, varying in socioeconomic status and spread over different neighborhoods in Amsterdam (The Netherlands). An a priori sample size calculation was done using Gpower 3.1 (Windows, Düsseldorf, Germany) [29]. With a power estimate of 0.80, alpha set at 0.05 and an effect size of 0.20, this led to a required sample size of 193. Taking into account expected exclusions, we aimed at including approximately 250 children per group and therefore 6 schools. Based on postal codes, three schools were located in an area with low socioeconomic status, two in medium and one was located in an area with high socioeconomic status. Parents gave written informed consent for their children's participation in the study. The study protocol received written approval by the Ethics Committee of Tilburg University (EC-2019.72).

### Participants

All children of these schools were invited to participate in the study. Data was collected in a final sample of 2068 children (age 4–13, 50,6% boys). In these children, at least one test was performed. Descriptive statistics on the study sample can be found in Table 1. Reasons for exclusion were absence, injury and absence of informed consent.

### Instruments

**Motor skill competence.** Motor skill competence was assessed using the 4-Skills Scan [30]. This test is easy to conduct in a school setting and has been found to be both reliable (ICC = 0.93 for test–retest reliability and ICC = .97 for inter-rater reliability) [31] and valid (r = 0,58) [32]. The 4-Skills Scan consists of four components: 1. Jumping force (locomotion), 2. Bouncing ball (object control), 3. Standing still (stability) and 4. Jumping coordination

**Table 1. Descriptive statistics of the study sample.**

| Age Group | Sample size (N) | Sexe (n, % boys—girls) | BMI (mean (SD)) | n | MVPA[a], min/wk (mean (SD)) | n | Perceived motor competence (mean (SD)) | n | Motor Age (mean (SD)) | n |
|---|---|---|---|---|---|---|---|---|---|---|
| Age 4 | 135 | 74–61 (54,8–45,2%) | 15,48 (1,31) | 123 | 42,19 (51,60) | 32 | 3,22 (0,58) | 118 | 4,33 (0,80) | 117 |
| Age 5 | 266 | 144–122 (54,1–45,9%) | 15,59 (1,48) | 248 | 46,02 (53,62) | 88 | 3,27 (0,53) | 245 | 5,16 (0,85) | 248 |
| Age 6 | 286 | 142–144 (49,7–50,3%) | 15,48 (1,58) | 268 | 111,14 (92,65) | 105 | 3,37 (0,46) | 277 | 6,26 (1,08) | 266 |
| Age 7 | 268 | 143–125 (53,4–46,6%) | 15,83 (2,07) | 255 | 110,26 (80,47) | 114 | 3,34 (0,41) | 259 | 7,57 (1,26) | 248 |
| Age 8 | 243 | 119–124 (49,0–51,0%) | 16,32 (2,11) | 232 | 129,25 (105,58) | 107 | 3,21 (0,52) | 237 | 8,54 (1,18) | 231 |
| Age 9 | 251 | 124–127 (49,4–50,6%) | 16,83 (3,05) | 242 | 153,51 (105,84) | 188 | 3,14 (0,58) | 230 | 9,50 (1,13) | 237 |
| Age 10 | 219 | 110–109 (50,2–49,8%) | 17,53 (2,92) | 202 | 164,31 (155,75) | 195 | 3,12 (0,57) | 190 | 10,15 (1,08) | 202 |
| Age 11 | 222 | 116–106 (52,3–47,7%) | 18,40 (3,85) | 198 | 180,88 (147,00) | 205 | 3,17 (0,56) | 185 | 10,65 (0,97) | 193 |
| Age 12+ | 178 | 74–104 (41,6–58,4%) | 19,12 (3,74) | 162 | 138,86 (129,37) | 167 | 3,17 (0,54) | 156 | 10,84 (0,88) | 160 |
| Total sample | 2068 | 1045–1022 (50,6–49,4%) | 16,63 (2,83) | 1930 | 137,09 (125,47) | 1201 | 3,23 (0,53) | 1897 | 8,09 (2,37) | 1902 |

[a] MVPA: moderate-to-vigorous physical activity.

(coordination). The subscales contain 11 elements of increasing difficulty. Each element represents a 'motor age': the age based on the depicted motor skill competence. For example, 6-year old children are expected to be able to skip. If a child successfully skips (and fails at subsequent elements), they score a motor age of 6. The mean of the four components forms a total score. Comparing motor age to calendar age leads to a score for 'motor lead', the final score used in our analyses. A positive motor lead value indicates that a child performs better than to be expected based on calendar age, a negative motor lead value indicates that a child performs lower than to be expected.

**Perceived motor competence.** For different age groups, different instruments were used to measure perceived motor competence. For children between 4 and 7, the different versions of the Pictorial Scale of Perceived Competence (PSPC) were used [33]. This scale contains 24 questions in 4 subscales (school competence, physical competence, social acceptance and maternal acceptance). For this study, the six questions of the subscale physical/motor competence were used ($\alpha = 0.55$) (see Appendix). For children aged 8 and older, a Dutch translation of the Self-Perception Profile for Children (SPPC) [34] was used (CBSK) [35]. This questionnaire contains 36 questions in 6 subscales. The six questions of the subscale Athletic Competence were used for this study ($\alpha = 0.70$, test-retest $r = 0.83$) and converted into a total score by adding up the scores and dividing them by 6. Both questionnaires are constructed in a similar way, making children choose between two types of children and asking: "who do you resemble the most?" For example: "Some children are very good in sports and physical education, but some children aren't very good in sports in physical education. Who do you resemble the most?"

**Physical activity.** Physical activity was measured using an adapted questionnaire based on the ENERGY-questionnaire [36]. Questions about home situation and the questions regarding energy intake, sedentary behavior and attitude towards physical activities were

excluded, since they exceeded the scope of this study. The amount of minutes of participation in organized sport per week was calculated as a measure of moderate-to-vigorous physical activity (MVPA). Since children under 10 years cannot accurately estimate their physical activity levels [37], the questionnaire was sent to the parents (online) for children younger than 10 years old. This was decided in agreement with experts in Amsterdam, who have a lot of experience in administering questionnaires to children.

**Health-related physical fitness.** The Eurofit test [38] was used as a measure of health-related physical fitness in children from 6 year old. This test included 8 test items: 1. Standing long jump 2. Bent arm hang 3. Sit and reach 4. 10x5m shuttle run 5. Plate tapping 6. Sit-ups 7. Handgrip strength 8. Shuttle run test. A composite score for overall fitness was calculated by converting raw scores to age-specific z-scores. Calculating a composite score for physical fitness is not common practice, but has been done before [39, 40]. In 6- and 7-year old children, plate tapping and the shuttle run test were not performed.

**Weight status.** Height was measured using a stadiometer, weight was measured using an analog scale. Height was rounded to the nearest half cm, weight was measured in kg with one decimal. Children were measured without wearing shoes. BMI was calculated by dividing weight (kg) by the square of the height (m) and was converted to z-scores using WHO's bmi-for-age tables [41].

## Procedures

Data collection took place in the physical education (PE) classes during school hours. Class started with a general introduction by the PE-teacher. Additional explanation and demonstration was given by the test conductors at the specific test item. To minimize the emphasis on measuring and to prevent children from watching each other, children were instructed to play PE-activities. The children were individually called to perform the specific test with the test conductor. All test conductors received training to ensure protocol compliance. Also, a supervisor was always present to observe and assure measurement quality and to organize the test-setting.

To perform all measurements, three PE-classes (approximately 45 minutes per class) were necessary. The 4-Skills Scan, in combination with body height and weight measurements, was administered in one PE-class. The tests was conducted by dividing the children in four groups. Approximately every eight minutes the groups rotated to the next activity and test. For the Eurofit test, the children were divided over two groups: one group participated in an activity with the PE-teacher, one group performed eight different tests; items 1 to 7 of the Eurofit test and the physical activity questionnaire that was completed on iPads (children ≥ 10 years). Halfway during class, the two groups switched. The shuttle run test, combined with the perceived motor competence measurements, also took one PE-class. While one group executed the shuttle run test, the other group filled in the questionnaire. For the perceived motor competence questionnaires, children were taken out of the PE-class to a more quiet place. The questionnaires started with a short introduction and an example. When children understood, either the test conductor read out the real questions to the children and filled in their answer on an iPad (PSPC) or the children could read the questions and fill in the answers by themselves (CBSK). While the PSPC was administered individually, the CBSK was administered in small groups of approximately four children.

## Data analyses

Data was pre-processed using R (v 4.0.3) [42]. Total scores for motor lead, perceived motor competence and physical fitness were calculated when 75% of the individual scores were

available. This was done by dividing the sum of the available scores by the number of available items. Then, for every age group, a correlation analysis was carried out in MPlus 7.4 [43] using a Full Information Maximum Likelihood (FIML) estimator to account for missing values. In all models, variances of, and covariances between variables were freely estimated, thus resulting in saturated models with $df$ = 0 and a perfect model fit. The complex procedure in Mplus was used to account for non-independence of observations due to cluster sampling (children nested within schools). Alpha level for significance was set at .05. To check for changes in the correlations over time, a Fisher's r to z transformation was done on the correlation coefficients that were significant. Then, the test-statistic z was calculated by $z = \frac{z_{r1} - z_{r2}}{se\,(z_{r1} - z_{r2})}$ where

$se_{(zr1-zr2)} = \sqrt{\left(\frac{1}{n_1 - 3} + \frac{1}{n_2 - 3}\right)}$. Z-values were compared for every relationship in all age groups.

## Results

The results are shown in Table 2 and Fig 1. Correlation coefficients varied from low to strong [44] and were negative for the relationships with BMI.

The Z-statistics revealed that the correlation coefficients do not gradually change over time. Instead, there seems to be a tipping point: either an association changes from nonsignificant to significant or the strength of that association grows stronger. A summary of the results is shown in Table 3 and Fig 2. Detailed tables are presented in the Supporting Information.

The data show an association between motor competence and physical activity in children from seven years old. The association between motor competence and perceived motor competence is first detected at five years old and this association seems to strengthen at the age of seven years old, although at ages eight and nine no association is found. Similarly, an association between physical activity and perceived motor competence is detected at eight years old, although at 9 and 10 years old no association is found. Physical fitness shows an association with motor competence and physical activity right from the moment we started measuring physical fitness. This association with motor competence increases in strength from the age of eight, while the association with physical activity grows stronger at the age of seven. Lastly, the data show that associations between BMI and the other four factors emerge one by one: at eight years old an association between BMI and physical activity and between BMI and physical fitness develops, at 9 years old an association between BMI and motor competence emerges and only at 11 years old an association between BMI and perceived motor competence arises. Interestingly, we also found a stable association between perceived motor competence and

**Table 2. Summary of the correlation analysis per age group.**

| Relation | Age 4 | Age 5 | Age 6 | Age 7 | Age 8 | Age 9 | Age 10 | Age 11 | Age 12+ |
|---|---|---|---|---|---|---|---|---|---|
| Motor competence–physical activity | 0,084 | 0,008 | 0,08 | 0,178* | 0,274* | 0,144 | 0,247* | 0,261* | 0,273* |
| Motor competence–perceived motor competence | 0,22 | 0,171* | 0,134* | 0,271* | 0,047 | 0,123 | 0,303* | 0,318* | 0,255* |
| Perceived motor competence–physical activity | -0,099 | 0,153 | 0,044 | 0,007 | 0,219* | 0,132 | 0,215 | 0,241* | 0,318* |
| Motor competence–physical fitness | N/A | N/A | 0,45* | 0,382* | 0,53* | 0,634* | 0,618* | 0,604* | 0,473* |
| Physical fitness–physical activity | N/A | N/A | 0,221* | 0,395* | 0,438* | 0,186 | 0,349* | 0,297* | 0,296* |
| BMI–motor competence | -0,078 | -0,008 | -0,061 | -0,078 | -0,155 | -0,318* | -0,322* | -0,397* | -0,19 |
| BMI–physical activity | 0,047 | 0,141 | -0,068 | 0,116 | -0,237* | -0,165 | -0,147* | -0,305* | -0,172* |
| BMI–perceived motor competence | 0,03 | 0,054 | 0,007 | 0,022 | -0,069 | -0,077 | -0,092 | -0,244* | -0,142* |
| BMI–physical fitness | N/A | N/A | -0,164 | -0,014 | -0,294* | -0,325* | -0,317* | -0,382* | -0,12 |

*correlation coefficients significant at p < 0,05.

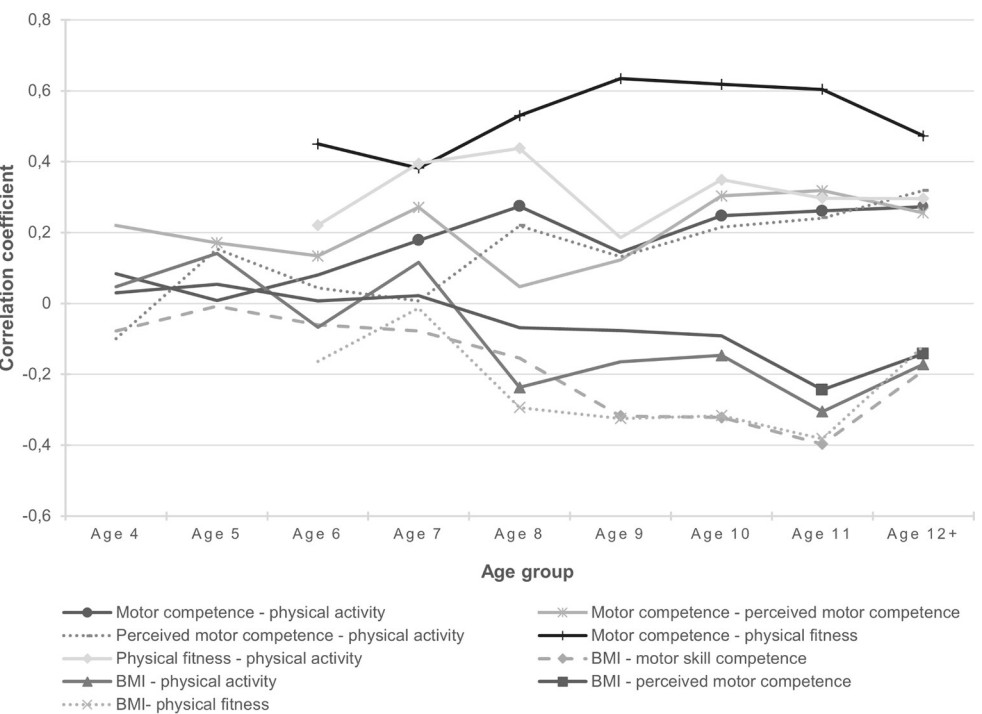

**Fig 1. Overview of the correlation coefficients of the different relationships in all age groups.** Nonsignificant correlations do not have a marker.

physical fitness right from the age where we started measuring physical fitness (6 yo.), although this relationship is not included in the model by Stodden et al. [15].

## Discussion

The goal of this study was to explore the relationships between weight status, physical activity, motor competence, perceived motor competence and physical fitness through time. To our knowledge, this is the first study including all five aspects included in the model by Stodden et al. [15] as well as a large sample of children of all ages from four to thirteen years old. Our findings show that all aspects are related to each other and that a tipping point exists at which relations between aspects emerge or at which they become stronger, confirming most of what is described in the developmental model by Stodden et al. [15].

**Table 3. Overview of the tipping points in the associations.**

| Relation | Type of tipping point | Age of tipping point |
|---|---|---|
| Motor competence–physical activity | No association → association | 6 to 7 yo. |
| Motor competence–perceived motor competence | Increased strength of association | 6 to 7 yo. |
| Perceived motor competence–physical activity | No association → association | 7 to 8 yo. |
| Motor competence–physical fitness | Increased strength of association | 7 to 8 yo. |
| Physical fitness–physical activity | Increased strength of association | 6 to 7 yo. |
| BMI–physical activity | No association → association | 7 to 8 yo. |
| BMI–physical fitness | No association → association | 7 to 8 yo. |
| BMI–motor competence | No association → association | 8 to 9 yo. |
| BMI–perceived motor competence | No association → association | 10 to 11 yo. |

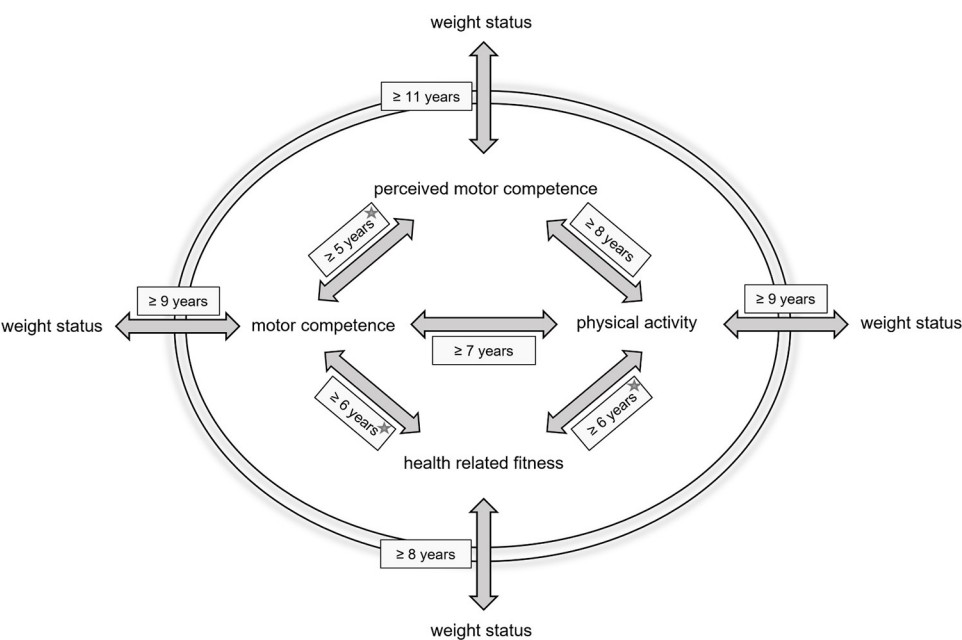

**Fig 2. The summary of the investigated age-dependent interrelations in model form.** The * indicates that the relationship increases in strength with age: at 7 for motor competence–perceived motor competence, at 8 for motor competence–fitness and at 7 for fitness–motor competence.

At the center of the model by Stodden et al. [15] is the relationship between motor competence and physical activity, that changes direction from early childhood to middle childhood. They propose that in early childhood, a weak relationship between motor competence and physical activity exists: physical activity stimulates the development of motor competence. We cannot confirm this proposed pathway, since we found no correlation between these two variables in 4- to 6-year old children. While Schmutz et al. [45] already found a weak relationship between motor competence and physical activity in early childhood, Nicolai Ré et al. [46] did not. In their review study, Barnett et al. [9] found no evidence of physical activity stimulating motor competence. Our results align with this conclusion, strengthening the notion that physical activity does not predict motor competence. As we have only measured time spent on physical activities, it cannot be ruled out that the quality of physical activity (i.e. amount of variation, free-play versus guided activities) does have a certain influence on the development of motor competence. Our data show that a relationship develops in middle childhood, which is in line with King-Dowling et al. [47] who also found that this relationship emerged over the study period. Since in multiple studies, including those in young kids (3 to 6 yo) [48–51], differences in physical activity were only found between high motor competence and moderate/low motor competence, it could be that a proficiency barrier exists [48] and that motor competence and physical activity will only be related in samples of children with higher average motor competence. In that case, children will only become more physically active when they have reached a certain level of proficiency in motor skills.

Perceived motor competence is described to be a mediator in the relationship between motor competence and physical activity: in early childhood perceived motor competence is on average high and stimulates both motor competence and physical activity, in middle childhood children develop the cognitive skills to accurately evaluate their own skills, at which point actual motor competence starts influencing perceived motor competence, influencing physical activity [15]. Our data show that children's perceived motor competence is already related to

actual motor competence at a young age, which is in line with other studies [52–54]. Although a recent review [55] was not able to demonstrate an age effect in the relationship between motor competence and perceived motor competence, our findings support the results of True et al. [56], showing the strengthening of the association between motor competence and perceived motor competence through developmental time. It therefore seems like motor competence of older children influence their perceived motor competence (and/or the other way around) more than in younger children demonstrate.

On the other hand, the proposed pathway from perceived motor competence to physical activity in early childhood cannot be confirmed in this study, since children up to 8 years old with higher perceived motor competence were not more physically active. This is in line with other cross-sectional studies in young children [52, 57–59]. Interestingly, both perceived and actual motor competence seem to not be related to physical activity behavior in young children. Our results show that perceived motor competence and physical activity are first related at 8 years old and that is also supported by other studies in samples of older children [26, 60, 61]. Hence, from this age on, perceived motor competence could have a mediating role as described in the developmental model [15]. This mediation could not yet be supported in a recent review [9]. While in this review study not enough studies were available to address changes with age, based on available studies age seems to be a factor, since all mediations that were found were in samples in which the children were 9 or older [9]. Therefore, it seems that whether young children feel like they are competent in performing motor skills or not, has no effect on the amount of time they spend on physical activities. When children grow older, children with higher perceived motor competence are also more physically active. This relationship is then equally strong as the direct relationship between actual motor competence and physical activity, which possibly underlines the importance of perceived motor competence and thereby the development of sufficient motor competence before perceived motor competence starts playing a role.

Although we only measured fitness from the age of six years old, our results confirm the notion that from a young age, physically fitter children are more competent in performing motor skills and are more physically active. These relationships become stronger when children grow older. While Barnett et al. [9] report insufficient evidence for a relationship between physical fitness and total motor competence and object control motor competence, they found strong evidence for reciprocal causal pathways between physical fitness and motor competence in locomotion/coordination/stability skills. Since our measure of motor competence only includes one object control skill and the other skills are in the domains of locomotion, coordination and stability, this could explain why our motor competence composite score is significantly related to physical fitness.

Finally, weight status is included as a product of all four factors [15]. We therefore included separate correlations between BMI and the other four factors in our analysis. We found no evidence that BMI was related to any of the other factors in the youngest children. Thereafter relationships appear gradually. The fact that BMI and motor competence were only related from the age of nine years old, does not concur with other studies looking at several age groups [28, 62, 63]. Although our results do align with Khodaverdi et al. [27] and Logan et al. [64], who also did not find that BMI and motor competence were related in young children, Logan et al. [64] did find a difference in motor competence between children with high BMI versus children with normal/low BMI. It could therefore still be possible that having high BMI negatively impacts motor skill development, but that within the range of low to normal BMI, differences in BMI are not related to differences in motor competence. Similarly, Khodaverdi et al. [27] also conclude that the low average BMI of their sample may have impacted their results.

Our findings also demonstrate that from the age of 11 children with higher BMI report lower perceived motor competence. Although an inverse relationship between BMI and

perceived motor competence was suggested in a recent review [65] not enough studies were available to confirm this, especially not in different age categories. This might therefore be a pathway that needs more research. Our findings show that children in late childhood who have higher perceived motor competence are also more physically active, have higher physical fitness, higher motor competence and lower BMI. Although this does not prove causality, it does align with the proposed spiral of (dis)engagement as described by Stodden et al. [15]. In middle childhood this relationship with BMI is not fully developed yet. Some longitudinal studies also support the proposed spiral. While motor competence influences fitness and fatness (directly and via fitness) from a young age [66, 67] and perceived motor competence later in childhood [68], motor competence [69, 70], perceived motor competence [68], fitness [23, 69] and weight [69] all appear to influence physical activity in late childhood. Our findings therefore reinforce the idea that focusing on motor competence in early childhood might be a feasible way to ensure continued participation in physical activities throughout childhood and adolescence and thereby reducing obesity problems.

Some relationships seem to weaken around the age of 12. This is the case for the relationship between BMI and motor competence, but also for the relationships of physical fitness with BMI, motor competence, and physical activity, where some relationships even disappear. Possibly, maturation plays an important role in the dynamic relationships between all these factors and other factors start becoming more important in the maintenance of BMI, physical activity, motor competence and fitness levels. Another possible explanation is that a ceiling effect exists for the 4 Skills Test, which might explain the weakening of relations with motor competence. Because of this, children can only perform below and on their expected motor age, not above, resulting in increased density of scores at the high end of the scale.

Multiple cross-sectional relationships proposed in the developmental model [15] have been studied together before, but only in specific age groups, mainly in older children [12, 23–27]. Our results largely concur with data from these studies, confirming relationships as proposed in the Stodden et al. [15] model. Our data extend these studies in the fact that we included a large sample of children from age 4 to 13, which made it possible to dive deeper into this model, exploring the age-dependency of the proposed relationships. Two studies [23, 26] found that fitness is a more important mediator than perceived motor competence. Indeed, our study also shows stronger associations between fitness and motor competence /physical activity than between perceived motor competence with motor competence / physical activity. However, this might also be explained by an overlap in content [71] and neuromuscular constraints [72] of tests of motor competence and physical fitness. In addition, Stodden et al. [40] pointed out that perceived motor competence may also play a role in the relation between motor competence and fitness, demonstrating another indirect pathway enforcing the spiral of (dis)engagement in physical activity. Our cross-sectional data show that there is indeed also an association between perceived motor competence and physical fitness in all age categories.

Some methodological issues seem to have impacted our results. No relation was found between motor competence and perceived motor competence at age eight and nine years old. This might be due to the fact that we changed to a different instrument (while both developed by the same author) for measuring perceived motor competence at eight years old. For young children we used a pictorial scale showing specific motor skills that were sometimes comparable with the motor skills tested by the 4 Skills Test. From 8 years old, children received a textual questionnaire, describing more generic performance at physical activities. Perhaps these questions were still too abstract for 8 and 9 year-olds, which could explain why no association was found between motor competence and perceived motor competence. Similarly, a sudden drop is seen in the relations with physical activity at the age of nine. This could be due to the fact that while we established that a self-report physical activity questionnaire was only suitable

from 10 years old, part of the 9 year-olds filled in the questionnaire themselves, instead of their parents. This was done in concurrence with the PE-teacher, who predicted no response from parents. This leads to the next limitation of this study, which is the lower response rate for the physical activity questionnaire that was sent to the parents. Although this was expected, it possibly led to bias in the data and led to a significantly lower sample size for the analyses that included physical activity in the younger children. Since it is also in the younger children that we found no relationships with physical activity, some caution is warranted there. In addition, assessing physical activity by use of a questionnaire often leads to an overestimation of physical activity [73]. Another limitation of this study is the cross-sectional nature of this study, preventing us from determining causal relationships. Strengths of this study are the use of the full information maximum likelihood procedure, the large sample of children between 4 and 13 years old and the inclusion of all five factors from the developmental model proposed by Stodden et al. [15]. This made it possible to look at the existence and strength of relationships in many age groups, using the same or similar instruments. However, the large scale of this study made it impossible to also look a separate aspects of the factors, which might be a necessary follow-up step. For example, it is argued that when studying the relationship between physical activity and motor competence, a distinction should be made between organized and non-organized physical activities [74]. For example, throwing and jumping skills were related to higher intensity, skill-specific physical activity after school, but not to the general level of physical activity [75]. In addition, motor competence and perceived motor competence did not predict general physical activity during the school day, but did predict playground physical activity [76]. Similarly, it has been proposed that locomotor skills may not contribute to the opportunities to participate in physical activities to the extent that ball skills do [77, 78], especially during school lunchtime and recess breaks [78]. In addition, ball skills seem to affect perceived motor competence more than locomotor skills do [16, 76]. In this study we included only time spent in organized sports activities as a measure of physical activity. Motor competence was included as a composite score of 4 skills and similarly, a composite score was calculated for health related fitness.

In conclusion, all five factors included in the developmental model by Stodden et al. [15] are related and tipping points exist after which the relations emerge or strengthen. It should be kept in mind that this is a complex system, in which many other factors might have interrelations with the factors described in this model. However, our results indicate that targeting motor competence and perceived motor competence at a young age might be a feasible way to ensure continued participation in physical activities throughout childhood and adolescence. Yet, how to effectively influence motor competence is still largely unknown: both our data and available literature suggest that only increasing physical activity will not be enough [9], while maintaining a healthy weight could be a promising starting point to kick off a positive spiral [9, 67]. Future research should thus aim to unravel how to improve motor competence. In addition, large scale longitudinal studies including all variables and all age groups are necessary to gain more insight in the directions of these relationships through developmental time, ideally while making the distinction between different aspects within each variable. Moreover, the possibility of nonlinearity of these relationships should be further investigated, since non-linear relationships between physical activity and motor skills [79] and between BMI and motor coordination [80] have been described. Lastly, addressing sex differences in these relationships might also be interesting as it has been proposed that the mediating role of perceived motor competence might be stronger for girls than for boys [26] and that reciprocal relations between motor competence, endurance and fatness are dependent on sex [67]. Since boys and girls do not go through their maturational stages simultaneously [81], exploring sex differences in the developmental nature of the studied interrelations would be a valuable direction for follow-up studies.

## Supporting information

**S1 Table. Z statistic values of the difference in correlation coefficients of BMI—motor competence between ages.** * 1,96 > z > -1,96 is significant. * n/a refers to a correlation coefficient being absent because measurements were net performed in that age group. The–means that one or two of the correlation coefficients were not significant. In both situations, no calculations on the significance of the difference could be performed.
(DOCX)

**S2 Table. Z statistic values of the difference in correlation coefficients of BMI—perceived motor competence between ages.** * 1,96 > z > -1,96 is significant. * n/a refers to a correlation coefficient being absent because measurements were net performed in that age group. The–means that one or two of the correlation coefficients were not significant. In both situations, no calculations on the significance of the difference could be performed.
(DOCX)

**S3 Table. Z statistic values of the difference in correlation coefficients of BMI—physical activity between ages.** * 1,96 > z > -1,96 is significant. * n/a refers to a correlation coefficient being absent because measurements were net performed in that age group. The–means that one or two of the correlation coefficients were not significant. In both situations, no calculations on the significance of the difference could be performed.
(DOCX)

**S4 Table. Z statistic values of the difference in correlation coefficients of BMI—physical fitness between ages.** * 1,96 > z > -1,96 is significant. * n/a refers to a correlation coefficient being absent because measurements were net performed in that age group. The–means that one or two of the correlation coefficients were not significant. In both situations, no calculations on the significance of the difference could be performed.
(DOCX)

**S5 Table. Z statistic values of the difference in correlation coefficients of motor competence—perceived motor competence between ages.** * 1,96 > z > -1,96 is significant. * n/a refers to a correlation coefficient being absent because measurements were net performed in that age group. The–means that one or two of the correlation coefficients were not significant. In both situations, no calculations on the significance of the difference could be performed.
(DOCX)

**S6 Table. Z statistic values of the difference in correlation coefficients of motor competence—physical activity between ages.** * 1,96 > z > -1,96 is significant. * n/a refers to a correlation coefficient being absent because measurements were net performed in that age group. The–means that one or two of the correlation coefficients were not significant. In both situations, no calculations on the significance of the difference could be performed.
(DOCX)

**S7 Table. Z statistic values of the difference in correlation coefficients of motor competence—physical fitness between ages.** * 1,96 > z > -1,96 is significant. * n/a refers to a correlation coefficient being absent because measurements were net performed in that age group. The–means that one or two of the correlation coefficients were not significant. In both situations, no calculations on the significance of the difference could be performed.
(DOCX)

**S8 Table. Z statistic values of the difference in correlation coefficients of perceived motor competence -physical activity between ages.** * 1,96 > z > -1,96 is significant. * n/a refers to a

correlation coefficient being absent because measurements were net performed in that age group. The–means that one or two of the correlation coefficients were not significant. In both situations, no calculations on the significance of the difference could be performed.
(DOCX)

**S9 Table. Z statistic values of the difference in correlation coefficients of physical activity—physical fitness between ages.** * 1,96 > z > -1,96 is significant. * n/a refers to a correlation coefficient being absent because measurements were net performed in that age group. The–means that one or two of the correlation coefficients were not significant. In both situations, no calculations on the significance of the difference could be performed.
(DOCX)

**S10 Table. Z statistic values of the difference in correlation coefficients of perceived motor competence—physical fitness between ages.** * 1,96 > z > -1,96 is significant. * n/a refers to a correlation coefficient being absent because measurements were net performed in that age group. The–means that one or two of the correlation coefficients were not significant. In both situations, no calculations on the significance of the difference could be performed.
(DOCX)

## Acknowledgments

The authors would like to thank the schools, children and parents for their cooperation and participation in this study. The help of the test conductors in data collection is also kindly acknowledged. Special thanks goes to M.J.M.H. Delsing and R.J. den Uil for their contributions in respectively data-analyses and writing.

## Author Contributions

**Conceptualization:** Anne R. den Uil.

**Formal analysis:** Anne R. den Uil.

**Funding acquisition:** Anne R. den Uil.

**Investigation:** Anne R. den Uil, Ilse T. Kat.

**Methodology:** Anne R. den Uil, Ilse T. Kat.

**Project administration:** Anne R. den Uil, Ilse T. Kat.

**Software:** Anne R. den Uil.

**Supervision:** Mirka Janssen, Vincent Busch, Ron H. J. Scholte.

**Writing – original draft:** Anne R. den Uil.

**Writing – review & editing:** Anne R. den Uil, Mirka Janssen, Vincent Busch, Ilse T. Kat, Ron H. J. Scholte.

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
