## [Decision Letter · Decision Letter 0]

2 Jan 2023

PONE-D-22-31483The relationships between children’s motor competence, physical activity, perceived motor competence, physical fitness and weight status in relation to agePLOS ONE

Dear Dr. den Uil,

Thank you for submitting your manuscript to PLOS ONE. After careful consideration, we feel that it has merit but does not fully meet PLOS ONE’s publication criteria as it currently stands. Therefore, we invite you to submit a revised version of the manuscript that addresses the points raised during the review process.

 Please see the reviewers suggestions to revise your paper. 

We look forward to receiving your revised manuscript.

Kind regards,

Ender Senel, PhD

Academic Editor

PLOS ONE

4. We note that Figure 1 in your submission contain copyrighted images. All PLOS content is published under the Creative Commons Attribution License (CC BY 4.0), which means that the manuscript, images, and Supporting Information files will be freely available online, and any third party is permitted to access, download, copy, distribute, and use these materials in any way, even commercially, with proper attribution. For more information, see our copyright guidelines: http://journals.plos.org/plosone/s/licenses-and-copyright.

a) You may seek permission from the original copyright holder of Figure(s) [#] to publish the content specifically under the CC BY 4.0 license. 

Reviewers' comments:

Reviewer's Responses to Questions

**Comments to the Author**

1. Is the manuscript technically sound, and do the data support the conclusions?

Reviewer #1: Partly

Reviewer #2: Yes

2. Has the statistical analysis been performed appropriately and rigorously? 

Reviewer #1: No

Reviewer #2: Yes

3. Have the authors made all data underlying the findings in their manuscript fully available?

Reviewer #1: Yes

Reviewer #2: Yes

4. Is the manuscript presented in an intelligible fashion and written in standard English?

Reviewer #1: Yes

Reviewer #2: Yes

5. Review Comments to the Author

Reviewer #1: This large-scale cross-sectional study examined correlations among five variables noted in the Stodden et al., conceptual model in children from 4-13 yrs, noting changes (or lack thereof) in the strength of correlations each year. The writing is generally clear and to the point.

This is the largest study to date that addresses all aspects of the model, which is a strength of the study. Specifically addressing the notion of hypothesized changes in the strength of associations across childhood and into adolescence is important as most studies in this area have not addressed this critical aspect of the model hypotheses and have not covered this wide of an age span.

While the cross-sectional nature of the data limits the generalizability of the conclusions, the data generally support other longitudinal data from studies that have examined a limited number of variables in the model. It might be useful to address the results of these studies to corroborate or refute the results of the current study as the age ranges of the below studies are within the age range of the current study.

Jaakkola, T., Yli‐Piipari, S., Huotari, P., Watt, A., & Liukkonen, J. (2016). Fundamental movement skills and physical fitness as predictors of physical activity: A 6‐year follow‐up study. Scandinavian Journal of Medicine & Science in Sports, 26(1), 74-81.

Lima, R.A., Pfeiffer, K.A., Bugge, A., Møller, N.C., Andersen, L.B., Stodden, D.F. (2017). Motor competence and cardiorespiratory fitness have greater influence on body fatness than physical activity across time. Scand J Med Sci Sport, 1-10. https://doi.org/10.1111/sms.12850

Lima, R. A., Bugge, A., Ersbøll, A. K., Stodden, D. F., & Andersen, L. B. (2019). The longitudinal relationship between motor competence and measures of fatness and fitness from childhood into adolescence. Jornal de Pediatria, 95, 482-488.

While the individual correlations among variables at each age is important in its own right, I am left wondering if it would be possible to analyze the data at each age as a collective system (e.g., using SEM/path analyses) as it would provide understanding of whether the data collectively “fit” the Stodden et al model as a whole (i.e., as a more comprehensive system of individual factors of development). This would provide a stronger conceptual understanding (while still providing an understanding of the individual strength of correlation coefficients in the models) of the overall fit of the conceptual model across ages, which was the intent of including all the different variables that have, historically, been examined individually. If the ”fit” of the individual age models strengthen (or do not) across time, then the central research question would still be answered (in addition to examining how individual correlations changed across time). As can be seen from the suggested “tipping point” time frames in Table 3, potentially demonstrating a non-significant model fit in younger ages would still address the central research question and account for the original hypotheses of the model, which suggests correlations among variables in the model (and the overall model fit) would be weaker in early childhood. However, based on the range of sample sizes at each age, and the number of variables that would need to be entered into each model, I am not sure if this suggestion is feasible from a statistical standpoint.

Another potential limitation of the current statistical analyses is the assumption of linear relationships between variables. The authors address this idea indirectly when referencing a proficiency barrier, but it is still a potential avenue for exploration, perhaps in a subsequent paper.

While the “motor age” variable partially addresses how motor skill scores generally increase with each age group, would it be useful to provide supplementary data to see the changes in raw motor skill scores across age groups? The authors noted a potential ceiling effect for the motor skill measures; thus, it might be useful to provide the raw data to better show the how motor skill levels change across time.

One important limitation in the data is based on the measure of PA. I believe it is important to note the limitations of these data more concretely. Specifically, the overestimation of PA with questionnaires should be noted.

Lastly, addressing maturation and how that impact gender-specific differences in the relationships (specifically during the adolescent transition) also is an important notion that was not explored in the data. Controlling for gender in the correlations might be a useful endeavor from a statistical standpoint.

Reviewer #2: Review_PONE-D-22-31483

Overview

The manuscript is excellent, the subject matter is current, and it is straightforward and objective. The research covers a significant information gap about the association between motor competence, physical activity, and related factors by using a well-designed approach and a large sample. Although I provide some suggestions for the authors' consideration, I firmly recommend publication of this article.

Introduction

Update the reference [2]

Materials and Method

• Give more information about the socioeconomic status of the sample or the population from which the sample was drawn.

• Give more details of excluded participants (exclusion percentage, gender, age group)

• Give more details on how the sampling was done; explain how the sample size was estimated.

Discussion

• Lines 300-302 - “Although we cannot draw conclusions on causality, our findings do not support the proposed pathway in which physical activity stimulates motor competence in early childhood.”

This sentence, unfortunately, is poorly constructed and may lead the reader into a misunderstanding. The authors are asked to consider rewriting it, bearing in mind that the data in this article preclude any stimulus/cause-effect inference. The choice of terms here must be very careful.

It is also suggested that the authors reflect a little more on the results of physical activity and possible biases in data collection, as it is precisely at younger ages that n is the lowest for the physical activity variable.

• Line 302-305: at this point, it should be considered that the present study used indirect measures of physical activity;

6. PLOS authors have the option to publish the peer review history of their article (what does this mean?). If published, this will include your full peer review and any attached files.

Reviewer #1: No

Reviewer #2: **Yes: **Maria Teresa Cattuzzo

---

## [Author Response · Author response to Decision Letter 0]

24 Feb 2023

Dear sir/madam,

Thank you for your review. We have addressed all question and remarks in our revised Cover Letter and Response to the Reviewers and have adjusted our Manuscript. We thereby hope our revised submission meets the requirements for publishing. 

Sincerely,

AR den Uil

---

## [Decision Letter · Decision Letter 1]

21 Mar 2023

The relationships between children’s motor competence, physical activity, perceived motor competence, physical fitness and weight status in relation to age

PONE-D-22-31483R1

Dear Dr. den Uil,

We’re pleased to inform you that your manuscript has been judged scientifically suitable for publication and will be formally accepted for publication once it meets all outstanding technical requirements.

Kind regards,

Ender Senel, PhD

Academic Editor

PLOS ONE

Additional Editor Comments (optional):

Reviewers' comments:

Reviewer's Responses to Questions

**Comments to the Author**

1. If the authors have adequately addressed your comments raised in a previous round of review and you feel that this manuscript is now acceptable for publication, you may indicate that here to bypass the “Comments to the Author” section, enter your conflict of interest statement in the “Confidential to Editor” section, and submit your "Accept" recommendation.

Reviewer #1: All comments have been addressed

2. Is the manuscript technically sound, and do the data support the conclusions?

Reviewer #1: Yes

3. Has the statistical analysis been performed appropriately and rigorously? 

Reviewer #1: Yes

4. Have the authors made all data underlying the findings in their manuscript fully available?

Reviewer #1: Yes

5. Is the manuscript presented in an intelligible fashion and written in standard English?

Reviewer #1: Yes

6. Review Comments to the Author

Reviewer #1: The authors have adequately addressed all my suggestions.

A suggestion for future work with these data... Cluster analyses... A different approach to address the data more from a person-centered (vs. a variable-centered) perspective.

7. PLOS authors have the option to publish the peer review history of their article (what does this mean?). If published, this will include your full peer review and any attached files.

Reviewer #1: No

---

## [Editor Report · Acceptance letter]

5 Apr 2023

PONE-D-22-31483R1 

The relationships between children’s motor competence, physical activity, perceived motor competence, physical fitness and weight status in relation to age 

Dear Dr. den Uil:

I'm pleased to inform you that your manuscript has been deemed suitable for publication in PLOS ONE. Congratulations! Your manuscript is now with our production department. 

Kind regards, 

on behalf of

Dr. Ender Senel 

Academic Editor

PLOS ONE